# Reverse Engineering Breast MRIs: Predicting Acquisition Parameters Directly from Images

**Nicholas Konz**[1]                              NICHOLAS.KONZ@DUKE.EDU

**Maciej A. Mazurowski**[1,2,3,4]           MACIEJ.MAZUROWSKI@DUKE.EDU

[1] *Department of Electrical and Computer Engineering*

[2] *Department of Radiology*

[3] *Department of Computer Science*

[4] *Department of Biostatistics & Bioinformatics*

*Duke University, NC, USA*

**Editors:** Accepted for publication at MIDL 2023

## Abstract

The image acquisition parameters (IAPs) used to create MRI scans are central to defining the appearance of the images. Deep learning models trained on data acquired using certain parameters might not generalize well to images acquired with different parameters. Being able to recover such parameters directly from an image could help determine whether a deep learning model is applicable, and could assist with data harmonization and/or domain adaptation. Here, we introduce a neural network model that can predict many complex IAPs used to generate an MR image with high accuracy solely using the image, with a single forward pass. These predicted parameters include field strength, echo and repetition times, acquisition matrix, scanner model, scan options, and others. Even challenging parameters such as contrast agent type can be predicted with good accuracy. We perform a variety of experiments and analyses of our model's ability to predict IAPs on many MRI scans of new patients, and demonstrate its usage in a realistic application. Predicting IAPs from the images is an important step toward better understanding the relationship between image appearance and IAPs. This in turn will advance the understanding of many concepts related to the generalizability of neural network models on medical images, including domain shift, domain adaptation, and data harmonization.

**Keywords:** MRI, acquisition parameters, breast, domain shift, inverse problem

## 1. Introduction

Magnetic resonance images (MRIs) are obtained under many different settings, such as the field strength, repetition and echo time, the type of contrast agent used, the acquisition matrix, as well as simply the scanner manufacturer and model (Chua et al., 2015; Saha et al., 2018). We collectively label these settings as the *image acquisition parameters* (IAPs). MRI scans are often used to train neural networks for medical image analysis tasks (Liu et al., 2018a; Angulakshmi and Deepa, 2021), such as breast cancer detection (Saha et al., 2018). While deep approaches often provide superior performance compared to early methods in medical image analysis (Le et al., 2019), they are also more prone to overfitting and *domain shift*. Domain shift is a problem in deep learning where the distribution of the data used to train a model differs from the distribution of the data used to test the model, and overfitting to the training domain(s) results in poor generalization to the test domain(s) (Guan and Liu, 2021; Wang and Deng, 2018).

In the context of medical imaging and MRIs in particular, domain shift occurs when images are taken according to different IAPs (along with other factors such as patient demographics and disease characteristics) (Glocker et al., 2019; Guan and Liu, 2021). Even if large training datasets are used that cover a variety of IAPs, it can still be common for datasets to only represent a fraction of all possible MRI IAP settings; *e.g.*, due to an institution using only certain scanner types, which can cause model generalization issues.

Due to the susceptibility of neural networks to domain shift, it is essential to understand the precise domain/IAPs of data before using it to train or test a model. For example, knowing the IAPs of the data where the model will be applied allows developers to determine whether the target scans (1) fall in the range of training data and the model can be applied with more confidence or (2) fall outside of that range and the model should be applied with more caution (or not applied). However, IAPs are not always recorded and available in datasets. If the precise domain/IAPs of data *is* known, the development of image harmonization/standardization or domain adaptation techniques (Sec. 2) for transforming images and/or models to a particular desired domain becomes possible.

In this paper, we present a model (Fig. 1) that solves this *inverse problem* of recovering the IAPs that created an MR image using only the image itself.

**Contributions.** Our contributions are summarized as follows.

1. We introduce a neural network model for predicting many categorical and continuous IAPs of an MR image in one forward pass, trained via multi-task learning (Sec. 4.1).

2. We show that our model predicts many complex IAPs of MRI scans of new patients to high accuracy, over a test set of about $2,000$ slice images, with a series of experiments (Sec. 5.1). We predict six out of ten categorical IAPs to $> 97\%$ top-1 accuracy on the test set, and all but two with $> 95\%$ top-2 accuracy.

3. We show that our method achieves fair accuracy ($> 84\%$ top-1 accuracy, $> 95\%$ top-2) on IAPs that are more challenging to predict, such as contrast agent type.

4. We demonstrate a realistic application of our model: using it to sort new unlabeled data into different domains to determine which models to apply to the data for a downstream task (Sec. 5.2).

By showing that it is possible to learn a mapping between an image and it's domain-defining acquisition parameters, we take an important first step in precisely defining the relationship between medical images and their domain, laying the groundwork for future work on domain adaptation and image harmonization methods. All code and instructions for reproducing our results are available at https://github.com/mazurowski-lab/MRI-IAP-prediction.

## 2. Related Works

**Inverse problems and recovering imaging parameters from images.** The task of recovering the parameters that generated an image directly from the image can be thought of as a type of inverse problem (Ren et al., 2020), where the goal is to recover the input to a function from the output of that function. Another inverse problem in MRI is that of recovering the physical phenomena that generated an image, such as the spin proton

density (Benning and Ehrhardt, 2016), which is different from our task of recovering the scanner settings and characteristics that generated the image. This was attempted once for natural photographs (Laurence and Murphy, 2018), but was not successful.

**The domain shift problem and domain adaptation approaches.** The domain shift problem is well-established in both general computer vision (Wang and Deng, 2018) and medical image analysis, the latter of which is due to differing IAPs and other factors (Glocker et al., 2019). Common *domain adaptation* techniques for mitigating this phenomena in medical imaging include adapting a trained model for the target domain (or preventing drastic overfitting to the source domain) (Zakazov et al., 2021), or transforming test images to a common domain prior to some downstream task (Koch et al., 2022; Diao et al., 2022; Zhang et al., 2018; Modanwal et al., 2021). Other methods seek to encode the intrinsic information content of medical images (Konz et al., 2022) in a way that is invariant to domain shift (Wolleb et al., 2022; Yang et al., 2019; Cao et al., 2022; Sun et al., 2022).

## 3. Dataset and MRI Acquisition Parameters

We use the Duke Breast Cancer (DBC) MRI dataset (Saha et al., 2018) for all experiments, which contains dynamic contrast-enhanced MRIs of 922 biopsy-confirmed breast cancer patients from over a decade. A large amount of clinical, demographic, pathological, genomic, and other data is provided for each scan, including 12 various MRI acquisition parameters (IAPs) which we study in this paper. These IAPs can each take different values (either categorical or continuous), and are summarized in Table 1. The categorical IAPs range from having 2 to 27 categories/classes each.

For all experiments, we use a subset of 14,000 2D slices randomly sampled from the original dataset's 3D fat saturated scan volumes, split randomly by patient into training, validation and testing subsets of $9,952$; $2,064$; and $1,984$ images, respectively, ensuring that slices from a given patient only appeared in one subset. We provide statistics for IAPs (distributions, correlations, and overlap of combinations) present in each of these subsets in Appendix A.1. We excluded 5 of the 922 patients from our data due to missing IAP values.

## 4. Methods

### 4.1. IAP Prediction Model

Our goal is to train a neural network to take new breast MRI slices as input and predict the values of the IAPs that generated these images. We thought it more practical and scalable to use a single network/forward pass to predict all IAPs at once, rather than train a seperate model for each IAP. To do so, we used a ResNet-18 (He et al., 2016) modified so that the final fully-connected layer is used to predict all IAPs at once, described as follows.

Each of the categorical studied IAPs (1-10 in Table 1) is converted into a one-hot encoding problem with $C_k$ ($k = 1, \ldots, K$, $K = 10$) possible values/categories (Table 1), e.g., $C_2 = 8$ for *Scanner Model*. Each $k^{th}$ categorical IAP is classified according to the maximum of the $C_k$ corresponding units in the final layer. For the $M = 2$ continuous-valued IAPs TE and TR (in milliseconds), we use a single unit for each of these in our network output layer to predict the IAP value directly. As such, the final layer of our model

has $\sum_{i=1}^{K} C_i + 2$ units, where $\sum_{i=1}^{K} C_i$ is the total of the number of possible values for each of the 10 categorical IAPs. We summarize our model in Fig. 1.

To train our model, we take a multi-task learning approach similar to that of (Liu et al., 2018b). We use a loss function that combines cross-entropy losses for the categorical IAPs with mean squared error (MSE) losses for the continuous IAPs, of

$$\mathcal{L}_{\text{IAP}} = \lambda \sum_{k=1}^{K} \mathcal{L}_{\text{CE}}(\hat{y}_k, y_k) + \eta \sum_{m=1}^{M} \mathcal{L}_{\text{MSE}}(\hat{y}_m, y_m). \tag{1}$$

The first term in the loss is the sum of the classification losses for the categorical IAPs, where $\mathcal{L}_{\text{CE}}$ is the cross-entropy loss for the model's predicted class $\hat{y}_k$ and true class label $y_k$ for the $k^{th}$ categorical IAP. Similarly, the second term is the sum of the regression losses for the continuous IAPs, where $\mathcal{L}_{\text{MSE}}$ is the MSE loss for the $m^{th}$ continuous IAP with predicted value $\hat{y}_m$ and true value $y_m$. $\lambda$ and $\eta$ are hyperparameters that control the relative importance of the categorical and continuous losses, respectively. We use $\lambda = 1$ and $\eta = 1$ for all experiments following experimentation on the validation set.

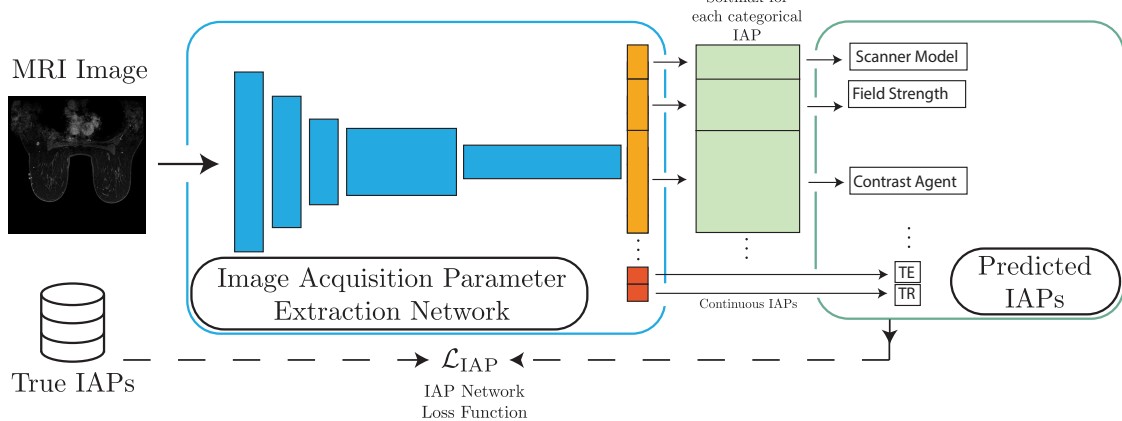

Figure 1: **Our MR image Acquisition Parameter (IAP) Extraction Model.** The model (Sec. 4.1) takes a breast MRI slice image as input, and predicts groups of class probabilities for each categorical IAP, and values for each continuous IAP (Table 1). The training pipeline is shown with dashed lines.

## 4.2. Experimental Settings

Our model was trained with a batch size of 512 for 100 epochs, using the Adam optimizer (Kingma and Ba, 2014) with a learning rate of 0.001 and weight decay strength of 0.0001. A model was saved for test set inference according to best performance on the validation set. All images were resized to 224×224 and normalized to $[0, 255]$, and all experiments were performed on a 48 GB NVIDIA A6000. The network took about 40 minutes to train.

## 5. Experiments and Results

We will first describe our experiments of using our model to predict the values of 12 IAPs (Table 1) of breast MR images of the test set of new patients (Sec. 5.1). Next, in Sec. 5.2 we will demonstrate an application of our IAP prediction model: using it to sort new unlabeled data into different domains to determine which trained models to apply to the data for a cancer classification task.

### 5.1. Predicting the acquisition parameters of MR images

In this section, we will describe our experiments of using our model to predict the values of 12 IAPs (Table 1) of breast MR images of the test set of new patients. We evaluated the performance of the network using mean squared error (MSE) for the continuous IAPs, and top-1 and top-2 accuracy for the categorical IAPs.

All IAP prediction results are shown in Table 1. We note again that all IAPs are predicted at the same time in a single forward pass of our model given an image, rather than separate passes/models for each IAP. We see that our model can predict most of the categorical IAPs to high accuracy, with $> 97\%$ top-1 accuracy on the test set, and all but two IAPs to greater than $> 95\%$ top-2 accuracy. Our model predicted the continuous IAPs (TE and TR) with relative errors of $< 1\%$ on average, as the TR values in the dataset range from 3.54–7.40ms, while TE ranges from 1.25–2.76ms. We also provide example predictions on specific diverse cases in Fig. 2. We see that our model is able to generalize to new patients taken with a variety of IAPs, able to predict the correct values for most of the IAPs; this is supported by the fact that the majority of unique IAP combinations of images in the test set are not present in the training set (Appendix A.1.2).

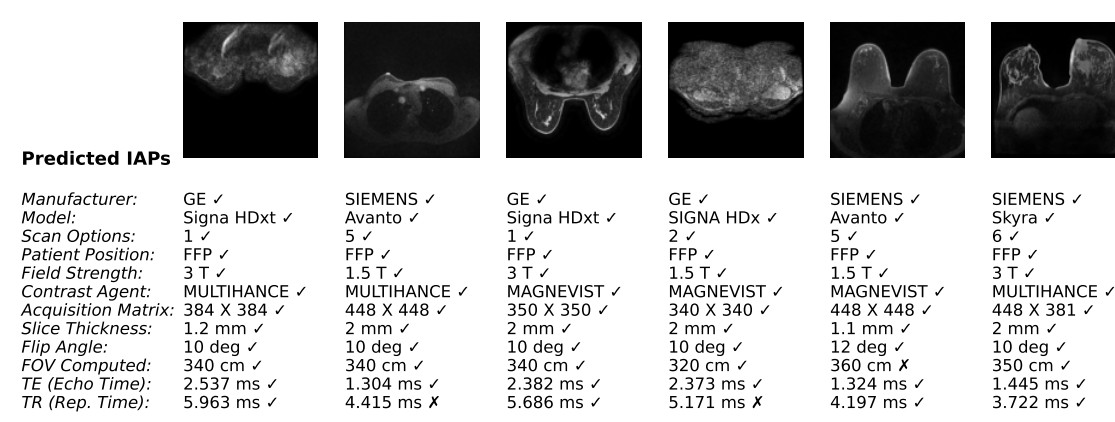

**Predicted IAPs**

| | | | | | | |
|---|---|---|---|---|---|---|
| *Manufacturer:* | GE ✓ | SIEMENS ✓ | GE ✓ | GE ✓ | SIEMENS ✓ | SIEMENS ✓ |
| *Model:* | Signa HDxt ✓ | Avanto ✓ | Signa HDxt ✓ | SIGNA HDx ✓ | Avanto ✓ | Skyra ✓ |
| *Scan Options:* | 1 ✓ | 5 ✓ | 1 ✓ | 2 ✓ | 5 ✓ | 6 ✓ |
| *Patient Position:* | FFP ✓ | FFP ✓ | FFP ✓ | FFP ✓ | FFP ✓ | FFP ✓ |
| *Field Strength:* | 3 T ✓ | 1.5 T ✓ | 3 T ✓ | 1.5 T ✓ | 1.5 T ✓ | 3 T ✓ |
| *Contrast Agent:* | MULTIHANCE ✓ | MULTIHANCE ✓ | MAGNEVIST ✓ | MAGNEVIST ✓ | MAGNEVIST ✓ | MULTIHANCE ✓ |
| *Acquisition Matrix:* | 384 X 384 ✓ | 448 X 448 ✓ | 350 X 350 ✓ | 340 X 340 ✓ | 448 X 448 ✓ | 448 X 381 ✓ |
| *Slice Thickness:* | 1.2 mm ✓ | 2 mm ✓ | 2 mm ✓ | 2 mm ✓ | 1.1 mm ✓ | 2 mm ✓ |
| *Flip Angle:* | 10 deg ✓ | 10 deg ✓ | 10 deg ✓ | 10 deg ✓ | 12 deg ✓ | 10 deg ✓ |
| *FOV Computed:* | 340 cm ✓ | 340 cm ✓ | 340 cm ✓ | 320 cm ✓ | 360 cm ✗ | 350 cm ✓ |
| *TE (Echo Time):* | 2.537 ms ✓ | 1.304 ms ✓ | 2.382 ms ✓ | 2.373 ms ✓ | 1.324 ms ✓ | 1.445 ms ✓ |
| *TR (Rep. Time):* | 5.963 ms ✓ | 4.415 ms ✗ | 5.686 ms ✓ | 5.171 ms ✗ | 4.197 ms ✓ | 3.722 ms ✓ |

Figure 2: **Example Predictions of Acquisition Parameters for MRIs in the Test Set.** Each image is from a different patient, and below each image are the predicted values for each of its IAPs (listed to the left). The symbols "✓" and "✗" indicate correct and incorrect predictions, respectively (TE and TR predictions are treated as "correct" if the relative error is $< 2\%$).

Table 1: **Model Prediction Performance for all IAPs on the Test Set.** Shown are all MR image acquisition parameters (IAPs) that we analyze, the number of possible categories of each in the dataset with examples, and the performance of our model for predicting them on the test set. Top-1 and top-2 prediction accuracy is provided for the categorical IAPs (first 10 rows), and mean squared error is provided for the continuous IAPs (bottom two rows). ∗ denotes prediction MSEs for models with categorical IAPs trained instead as continuous.

|  | **MRI acquisition parameter (IAP)** | **No. categories** | **Examples** | **Top-1 pred. acc. (%)** | **Top-2 pred. acc. (%)** | **Pred. MSE** |
|---|---|---|---|---|---|---|
| 1 | Scanner Manufacturer | 2 | GE, Siemens | 99.74 | N/A | N/A |
| 2 | Scanner Model | 8 | Avanto, Signa HDx | 97.78 | 99.29 | N/A |
| 3 | Scan Options | 9 | PFP/FS, PFP/SFS | 99.40 | 99.60 | N/A |
| 4 | Field Strength | 5 | 1.5 T, 3 T | 98.19 | 99.70 | N/A |
| 5 | Patient Position | 2 | FFP, HFP | 97.73 | N/A | N/A |
| 6 | Contrast Agent Type | 6 | Gadavist, MultiHance | 84.73 | 95.46 | N/A |
| 7 | Acquisition Matrix | 10 | $448 \times 448$, $384 \times 360$ | 91.53 | 99.14 | N/A |
| 8 | Slice Thickness | 21 | 1.3 mm, 2 mm | 76.66 | 87.05 | 0.157 mm∗ |
| 9 | Flip Angle | 4 | 10°, 12° | 99.65 | 99.75 | 0.073°∗ |
| 10 | FOV Computed | 27 | 320 cm, 360 cm | 51.21 | 69.30 | 164 cm∗ |
| 11 | Repetition Time (TR) | N/A | 4.27 ms, 5.34 ms | N/A | N/A | 0.0305 ms |
| 12 | Echo Time (TE) | N/A | 2.4 ms, 1.5 ms | N/A | N/A | 0.0116 ms |

### 5.1.1. The most challenging acquisition parameters

The two most challenging IAPs were *Slice Thickness* (21 categories) and *FOV Computed* (27 cats.), which had noticeably lower accuracies than the other IAPs (76.66% and 51.21%, respectively). Both are physically continuous parameters that were discretized into many categories when the dataset was created, which is why the top-2 accuracies are much higher (87.05% and 69.30%, respectively), as our model was close to the correct values. We also provide MSE results for training a model to instead regress these two IAPs and *Flip Angle* as continuous parameters, in Table 1. *Contrast Agent Type* was also more difficult to predict than most other categorical IAPs (84.73% accuracy); this is because predicting this IAP from an image forms a difficult inverse problem, *i.e.*, the mapping between it and the final image is not necessarily one-to-one.

## 5.2. Application: Sorting Unlabeled Data into Domains for Downstream Task Model Selection

Here we will our model to mitigate domain shift by sorting new MR images without IAP labels into different domains, to determine which cancer binary classification models trained in different domains to apply to the data. We trained a ResNet-18 for this task on only images taken with a GE scanner from the training set, the "**GE Model**", and similar for the "**Siemens Model**", using the same training settings as before. Slices containing a tumor bounding box are *positive*, and slices $\geq 5$ slices away from the positives are *negative.* We used the same test set as before, which has both GE and Siemens images (Sec. 3).

Our results are summarized in Table 2. Without our IAP prediction model, the scanner manufacturers (GE or Siemens) of the test images remain unknown so the only way to proceed is to either use the GE Model or the Siemens model, and it is unknown if the correct model is being used for an image; this resulted in classification accuracies of 68.86% and 56.50%, respectively. However, with our trained IAP prediction model (Sec. 5.1), we can determine the correct model to apply to each image according to the predicted manufacturer of that image from the IAP model. This resulted in a cancer classification accuracy of **76.95%**, a significant improvement over the uninformed approach.

Table 2: **Using our IAP prediction model to sort unlabeled data for cancer classification model selection.** See Sec. 5.2; values shown are cancer classification accuracies on the test set of GE and Siemens images, unless otherwise stated.

| GE Model | GE Model (on only GE images) | Siemens Model | Siemens Model (on only Siemens images) | Model chosen according to predicted IAPs | Model chosen according to true IAPs |
|---|---|---|---|---|---|
| 68.82% | 80.37% | 56.50% | 71.75% | **76.95%** | 77.43% |

## 6. Discussion

In this paper, we introduced a model that can predict the image acquisition parameters (IAPs) that generated an MR image solely from the image. Trained via multi-task learning on a breast MRI dataset with 12 analyzed IAPs, our model was able to predict most IAPs of new patient scans in the test set to high accuracy. We also demonstrated the usefulness of our model for sorting unlabeled data into domains for cancer classification model selection.

We found very good performance ($> 97\%$ top-1 accuracy) on predicting the categorical IAPs of *Scanner Manufacturer* (2 categories), *Scanner Model* (8 cats.), *Scan Options* (9 cats.), *Field Strength* (5 cats.), *Patient Position* (2 cats.) and *Flip Angle* (4 cats.). *Acquisition Matrix* (10 cats.) was more challenging (91.53% top-1 acc.), but still good. The continuous IAPs of TE and TR were also predicted to high accuracy. Some of these IAPs seem trivial to predict (*e.g.*, *Patient Position*), but it is surprising that others such as *Scan Options* and TE/TR can be recovered, as these are not necessarily clear from the image.

For the IAPs mentioned so far, the mapping of IAP value to the final image is mostly predictable, and the domain change of the image due to the IAP is clear. This also confirms that indeed, the predictions of neural networks are clearly affected by the domain of the image, as these IAPs affect the domain and they can be predicted solely from the image.

*Contrast Agent Type* (84.73% top-1, 6 cats.), *Slice Thickness* (76.77% top-1, 21 cats.) and *FOV Computed* (51.21% top-1, 27 cats.) were the more challenging IAPs to predict (discussed in Sec. 5.1.1). However, our model was still able to perform much better than random guessing, especially when considering the much higher top-2 accuracies (95.46%, 87.05% and 69.30%, respectively). This means that the mapping of these IAPs to the final image is uncertain and not as predictable as the other IAPs, but still somewhat learnable.

There are certain limitations to our study. One is that the dataset has an imbalanced distribution of images with respect to certain IAP values (Appendix A.1), which could result in biased evaluation metrics. Correlations also exist between certain IAPs (App. A.1.1), so predicting some IAPs may be easier than others; however, our model's test performance goes far beyond what could be inferred solely from these correlations (especially *Contrast Agent Type*, for example, which has low corr. with other IAPs and yet we predicted to high accuracy). Finally, the generalizability our model is promising given that the majority of unique IAP combinations present in test images were unseen during training (App. A.1.2).

Possible future work could include improving our model using other multi-task learning methods, or using 2.5D or 3D convolutional nets on the full 3D scan volumes, which was beyond the scope of this "proof of concept" work. We would also like to test our model's ability to generalize to other MR datasets and/or modalities that include IAPs, such as brain MRI. Finally, applications of our model's ability to predict domain characteristics (IAPs) from images for domain adaptation or data harmonization methods should be explored.

## 7. Conclusion

We presented a neural network model for inferring the image acquisition parameters (IAPs) that generated an MR image solely from the image, and obtained highly accurate predictions for most IAPs on a breast MRI dataset. This lays the groundwork for a better understanding of the relationship between medical images and their domain, which could be useful for data harmonization methods, domain adaptation approaches, and other applications.

## Acknowledgments

Research reported in this publication was supported by the National Institute Of Biomedical Imaging And Bioengineering of the National Institutes of Health under Award Number R01EB031575. The content is solely the responsibility of the authors and does not necessarily represent the official views of the National Institutes of Health.

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

## Appendix A. Additional Dataset Details

### A.1. Distributions of IAP Values in Dataset

In Figures 3, 4 and 5 we provide histograms for the distribution of each IAP's values in the training, validation and test sets, respectively. IAP values are indicated by their category/class index for all except TE and TR (which have their millisecond values); please see columns B:S, rows 2:3 of the spreadsheet https://wiki.cancerimagingarchive.net/download/attachments/70226903/Clinical_and_Other_Features.xlsx for an explanation of what each IAP index value represents.

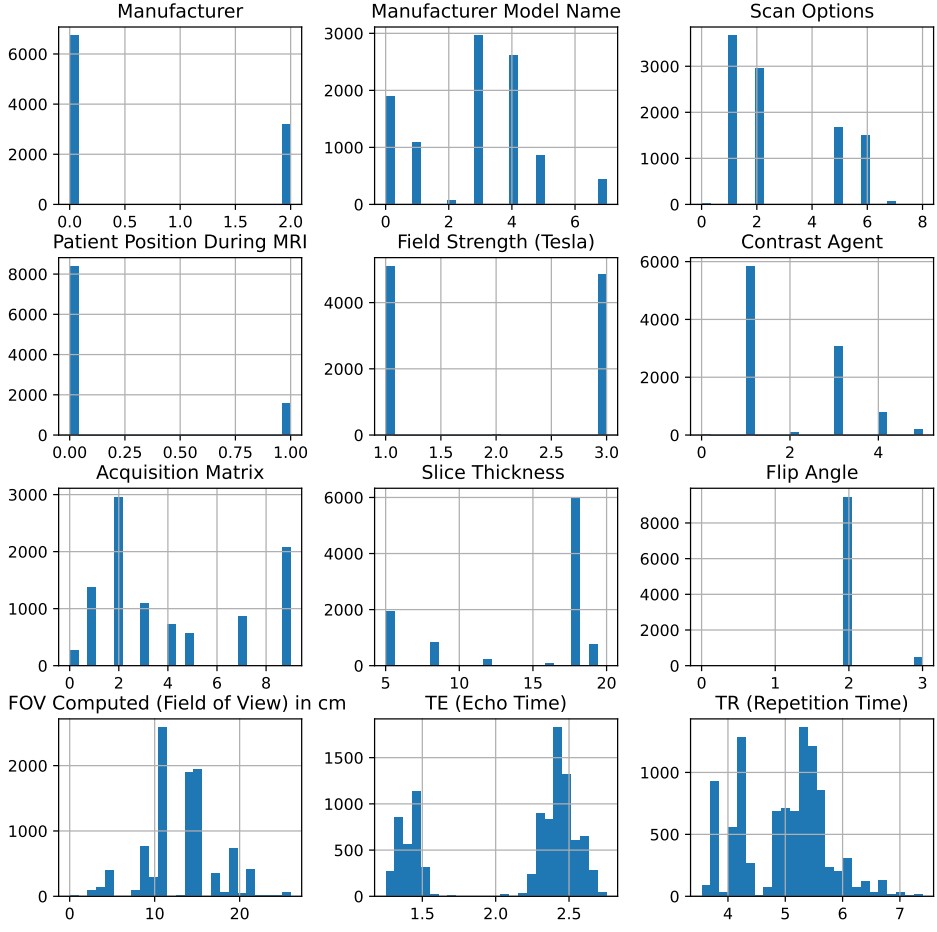

Figure 3: **Distribution of IAP values in the training set.**

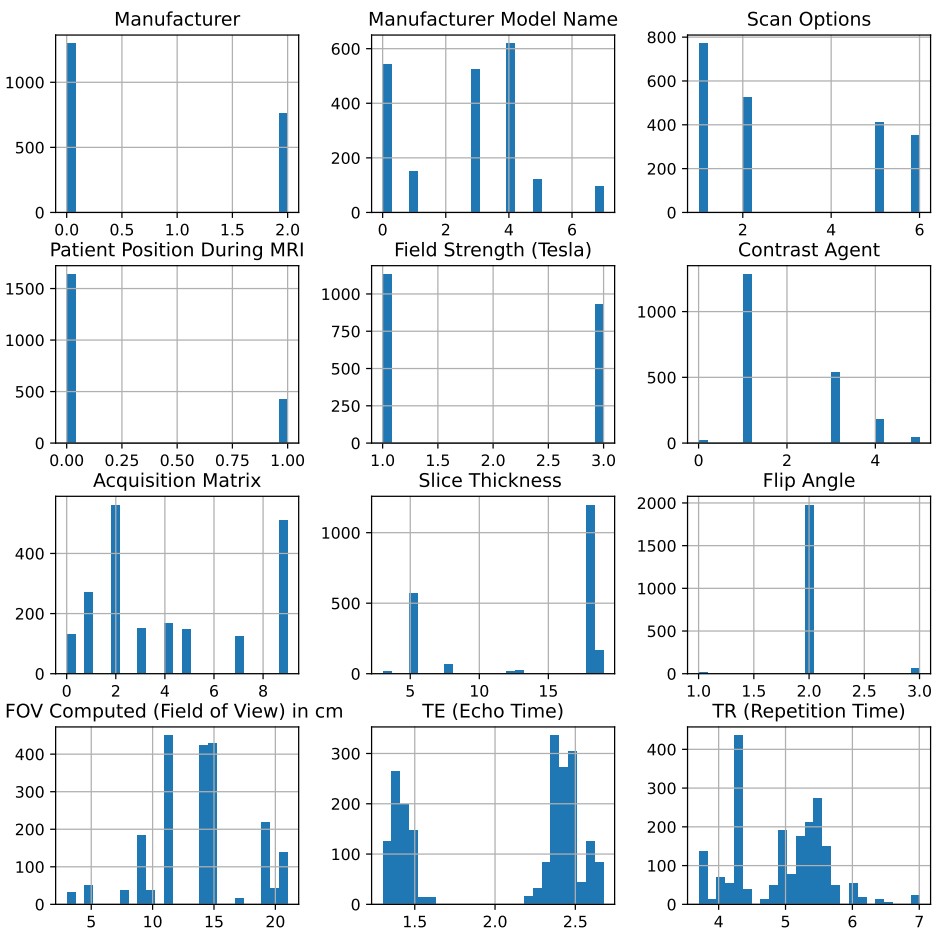

Figure 4: **Distribution of IAP values in the validation set.**

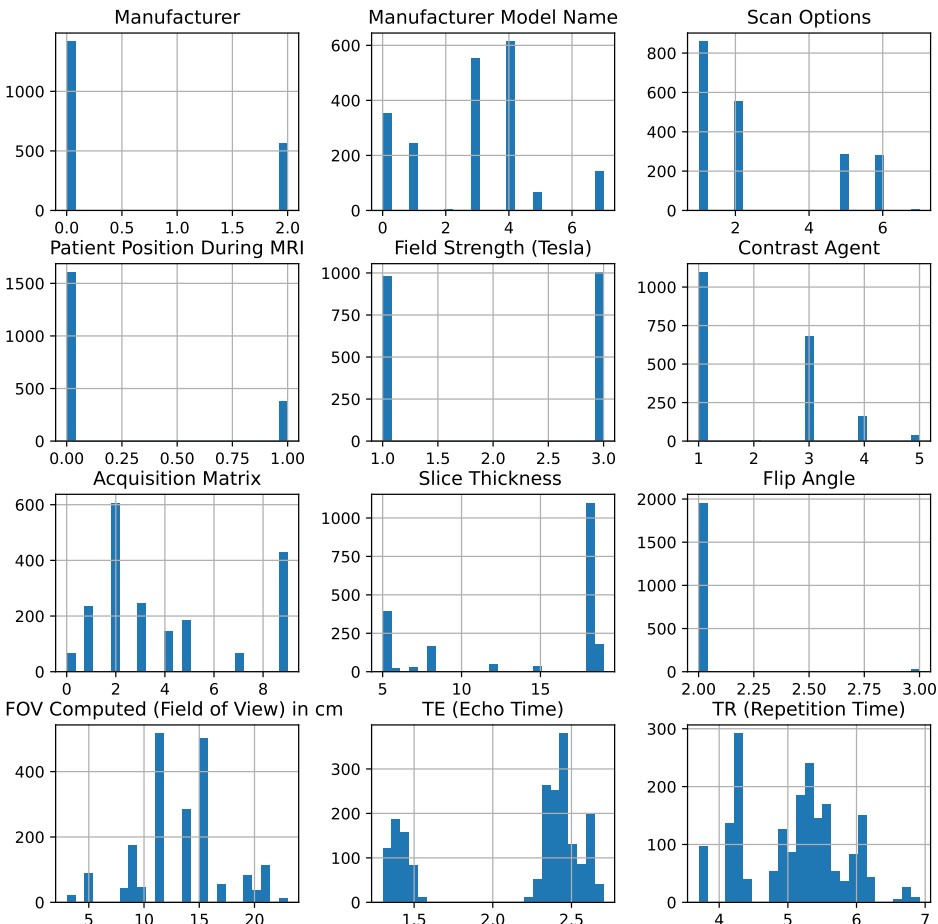

Figure 5: **Distribution of IAP values in the test set.**

### A.1.1. Correlations between IAPs

In Figures 6, 7, and 8 we provide the Spearman non-linear rank correlation coefficients between the different IAPs (Table 1) in the training, validation and test sets, respectively.

| | Manufacturer | Manufacturer Model Name | Scan Options | Patient Position During MRI | Field Strength (Tesla) | Contrast Agent | Acquisition Matrix | Slice Thickness | Flip Angle \n | FOV Computed (Field of View) in cm | TE (Echo Time) | TR (Repetition Time) |
|---|---|---|---|---|---|---|---|---|---|---|---|---|
| **Manufacturer** | 1.000000 | -0.181517 | 0.836731 | 0.651983 | -0.120854 | 0.122611 | 0.643939 | -0.331231 | 0.156226 | 0.051600 | -0.817374 | -0.817166 |
| **Manufacturer Model Name** | -0.181517 | 1.000000 | -0.153215 | -0.279388 | 0.524761 | 0.054491 | -0.054907 | 0.118374 | -0.255100 | 0.031841 | 0.215104 | 0.007772 |
| **Scan Options** | 0.836731 | -0.153215 | 1.000000 | 0.479051 | 0.174910 | -0.106335 | 0.374051 | -0.317145 | 0.166353 | -0.060150 | -0.723016 | -0.649471 |
| **Patient Position During MRI** | 0.651983 | -0.279388 | 0.479051 | 1.000000 | -0.200698 | 0.089797 | 0.602229 | -0.176787 | -0.055317 | -0.105688 | -0.596528 | -0.454946 |
| **Field Strength (Tesla)** | -0.120854 | 0.524761 | 0.174910 | -0.200698 | 1.000000 | -0.284033 | -0.022014 | -0.132017 | -0.118601 | -0.057580 | 0.189041 | 0.295728 |
| **Contrast Agent** | 0.122611 | 0.054491 | -0.106335 | 0.089797 | -0.284033 | 1.000000 | 0.453881 | 0.228686 | -0.096685 | 0.213296 | 0.116495 | -0.118733 |
| **Acquisition Matrix** | 0.643939 | -0.054907 | 0.374051 | 0.602229 | -0.022014 | 0.453881 | 1.000000 | -0.102878 | 0.154968 | 0.097385 | -0.353865 | -0.389098 |
| **Slice Thickness** | -0.331231 | 0.118374 | -0.317145 | -0.176787 | -0.132017 | 0.228686 | -0.102878 | 1.000000 | -0.249037 | -0.096434 | 0.273679 | 0.294226 |
| **Flip Angle \n** | 0.156226 | -0.255100 | 0.166353 | -0.055317 | -0.118601 | -0.096685 | 0.154968 | -0.249037 | 1.000000 | -0.099460 | -0.123969 | -0.106427 |
| **FOV Computed (Field of View) in cm** | 0.051600 | 0.031841 | -0.060150 | -0.105688 | -0.057580 | 0.213296 | 0.097385 | -0.096434 | -0.099460 | 1.000000 | -0.109579 | -0.276461 |
| **TE (Echo Time)** | -0.817374 | 0.215104 | -0.723016 | -0.596528 | 0.189041 | 0.116495 | -0.353865 | 0.273679 | -0.123969 | -0.109579 | 1.000000 | 0.851514 |
| **TR (Repetition Time)** | -0.817166 | 0.007772 | -0.649471 | -0.454946 | 0.295728 | -0.118733 | -0.389098 | 0.294226 | -0.106427 | -0.276461 | 0.851514 | 1.000000 |

Figure 6: **Spearman Correlations between IAPs for images in the training set.**

### A.1.2. IAP Combinations Present In Training, Validation and Testing Datasets

In this section, we will describe the numbers of unique combinations of IAP values that are present in the training, validation and testing datasets, and show how many of these combinations appear in which datasets. For example, we show the number of unique IAP combinations present in the training set but not the testing set, vice versa, and the number present in both. All of this information is in Table 3.

| | Manufacturer | Manufacturer Model Name | Scan Options | Patient Position During MRI | Field Strength (Tesla) | Contrast Agent | Acquisition Matrix | Slice Thickness | Flip Angle \n | FOV Computed (Field of View) in cm | TE (Echo Time) | TR (Repetition Time) |
|---|---|---|---|---|---|---|---|---|---|---|---|---|
| Manufacturer | 1.000000 | -0.356343 | 0.816917 | 0.762144 | -0.228361 | 0.099059 | 0.593022 | -0.212205 | 0.200270 | -0.228655 | -0.781115 | -0.780957 |
| Manufacturer Model Name | -0.356343 | 1.000000 | -0.338288 | -0.189119 | 0.538308 | -0.036706 | -0.100611 | 0.058776 | -0.260493 | 0.114297 | 0.369125 | 0.280946 |
| Scan Options | 0.816917 | -0.338288 | 1.000000 | 0.603996 | 0.040063 | -0.099021 | 0.311262 | -0.123485 | 0.180438 | -0.215974 | -0.735729 | -0.650727 |
| Patient Position During MRI | 0.762144 | -0.189119 | 0.603996 | 1.000000 | -0.123283 | 0.214215 | 0.651481 | -0.083965 | -0.034759 | -0.204838 | -0.624039 | -0.571512 |
| Field Strength (Tesla) | -0.228361 | 0.538308 | 0.040063 | -0.123283 | 1.000000 | -0.328982 | -0.041079 | -0.087075 | -0.134755 | -0.011673 | 0.252783 | 0.450979 |
| Contrast Agent | 0.099059 | -0.036706 | -0.099021 | 0.214215 | -0.328982 | 1.000000 | 0.490032 | 0.156180 | -0.039432 | 0.114176 | 0.232826 | -0.068317 |
| Acquisition Matrix | 0.593022 | -0.100611 | 0.311262 | 0.651481 | -0.041079 | 0.490032 | 1.000000 | -0.018346 | 0.166454 | -0.156887 | -0.178640 | -0.258974 |
| Slice Thickness | -0.212205 | 0.058776 | -0.123485 | -0.083965 | -0.087075 | 0.156180 | -0.018346 | 1.000000 | -0.219705 | -0.167078 | 0.199404 | 0.311315 |
| Flip Angle \n | 0.200270 | -0.260493 | 0.180438 | -0.034759 | -0.134755 | -0.039432 | 0.166454 | -0.219705 | 1.000000 | -0.191984 | -0.153330 | -0.175759 |
| FOV Computed (Field of View) in cm | -0.228655 | 0.114297 | -0.215974 | -0.204838 | -0.011673 | 0.114176 | -0.156887 | -0.167078 | -0.191984 | 1.000000 | 0.021513 | -0.146313 |
| TE (Echo Time) | -0.781115 | 0.369125 | -0.735729 | -0.624039 | 0.252783 | 0.232826 | -0.178640 | 0.199404 | -0.153330 | 0.021513 | 1.000000 | 0.843812 |
| TR (Repetition Time) | -0.780957 | 0.280946 | -0.650727 | -0.571512 | 0.450979 | -0.068317 | -0.258974 | 0.311315 | -0.175759 | -0.146313 | 0.843812 | 1.000000 |

Figure 7: **Spearman Correlations between IAPs for images in the validation set.**

| | Manufacturer | Manufacturer Model Name | Scan Options | Patient Position During MRI | Field Strength (Tesla) | Contrast Agent | Acquisition Matrix | Slice Thickness | Flip Angle \n | FOV Computed (Field of View) in cm | TE (Echo Time) | TR (Repetition Time) |
|---|---|---|---|---|---|---|---|---|---|---|---|---|
| Manufacturer | 1.000000 | -0.056716 | 0.854774 | 0.545802 | -0.073497 | 0.162256 | 0.789870 | -0.295707 | 0.369046 | -0.031436 | -0.828083 | -0.827905 |
| Manufacturer Model Name | -0.056716 | 1.000000 | -0.017454 | 0.117144 | 0.478684 | 0.037323 | -0.116088 | 0.164291 | -0.408908 | -0.195158 | 0.163255 | -0.122424 |
| Scan Options | 0.854774 | -0.017454 | 1.000000 | 0.482455 | 0.250981 | -0.026008 | 0.596903 | -0.299401 | 0.249645 | -0.010076 | -0.723460 | -0.713024 |
| Patient Position During MRI | 0.545802 | 0.117144 | 0.482455 | 1.000000 | 0.037235 | -0.052890 | 0.536498 | -0.243474 | -0.115344 | -0.354520 | -0.410325 | -0.413853 |
| Field Strength (Tesla) | -0.073497 | 0.478684 | 0.250981 | 0.037235 | 1.000000 | -0.229743 | -0.038453 | -0.116655 | -0.277382 | -0.068235 | 0.169700 | 0.198636 |
| Contrast Agent | 0.162256 | 0.037323 | -0.026008 | -0.052890 | -0.229743 | 1.000000 | 0.379308 | 0.377462 | -0.187635 | 0.257902 | 0.086627 | -0.101958 |
| Acquisition Matrix | 0.789870 | -0.116088 | 0.596903 | 0.536498 | -0.038453 | 0.379308 | 1.000000 | -0.237060 | 0.364072 | 0.020224 | -0.478861 | -0.504258 |
| Slice Thickness | -0.295707 | 0.164291 | -0.299401 | -0.243474 | -0.116655 | 0.377462 | -0.237060 | 1.000000 | -0.408606 | 0.069632 | 0.193404 | 0.225767 |
| Flip Angle \n | 0.369046 | -0.408908 | 0.249645 | -0.115344 | -0.277382 | -0.187635 | 0.364072 | -0.408606 | 1.000000 | -0.003147 | -0.365065 | -0.271836 |
| FOV Computed (Field of View) in cm | -0.031436 | -0.195158 | -0.010076 | -0.354520 | -0.068235 | 0.257902 | 0.020224 | 0.069632 | -0.003147 | 1.000000 | -0.059255 | -0.149036 |
| TE (Echo Time) | -0.828083 | 0.163255 | -0.723460 | -0.410325 | 0.169700 | 0.086627 | -0.478861 | 0.193404 | -0.365065 | -0.059255 | 1.000000 | 0.835238 |
| TR (Repetition Time) | -0.827905 | -0.122424 | -0.713024 | -0.413853 | 0.198636 | -0.101958 | -0.504258 | 0.225767 | -0.271836 | -0.149036 | 0.835238 | 1.000000 |

Figure 8: **Spearman Correlations between IAPs for images in the test set.**

Table 3: **Overlap of unique IAP combinations present in the training, validation and testing sets.** Each entry in the table is the number of unique combinations of IAPs for images in the described dataset.

| Subset $A$ | Subset $B$ | Num. in $A$ but not $B$ | Num. in $B$ but not $A$ | Num. in both $A$ and $B$ |
|---|---|---|---|---|
| Training | Validation | 385 | 61 | 44 |
| Training | Testing | 386 | 71 | 43 |
| Validation | Testing | 88 | 97 | 17 |

