# OpenReview forum: "Reverse Engineering Breast MRIs: Predicting Acquisition Parameters Directly from Images"
_MIDL.io/2023/Conference — MIDL 2023 Poster_

### Official Review · Reviewer_a8L3 · 2023-01-29

**Confidence:** 5
**Preliminary Rating:** 4
**Recommendation:** Poster

**Summary:**

This paper introduces a neural network model that can accurately predict image acquisition parameters (IAPs) from MRI scans with a single forward pass. These IAPs include field strength, echo and repetition times, acquisition matrix, scanner model, scan options and contrast agent type. The authors conducted experiments to test the accuracy of their model in predicting these parameters on many MRI scans of new patients. This research is important for understanding how deep learning models can be applied to medical images by better understanding the relationship between image appearance and IAPs as well as helping with domain shift, domain adaptation and data harmonization.

**Strengths:**

- the authors have developed a neural network model that can accurately predict image acquisition parameters from MRI scans with a single forward pass. This is an important step towards better understanding the relationship between image appearance and IAPs which could help in areas like domain shift, domain adaptation and data harmonization.
- they conducted experiments to test how well their model can predict these parameters on many MRI scans of new patients which demonstrates its potential value to the community.
- the paper is well written

**Weaknesses:**

- the method has been evaluated only on breast MRI. This is a very specific setup and findings might not generalise to other sequences.
- how does the acquisition coil setup influence the prediction? Can you predict the used coil if you had that data?
- some parameters seem trivial, e.g. patient position but others might be influenced by patient presentation. How do the indiviudal parameters correlate with downstream model choice in Table 2?


**Deanonymize Review:**

no

**Detailed Comments:**

I found no typos and the paper seems sound and reproducible. It seems like an interesting proof-of-concept study.

**Paper Type:**

validation/application paper

**Questions To Address In The Rebuttal:**

In their rebuttal, I would like the authors to address any potential limitations of their model and provide more detailed information on the experiments conducted, such as sample size, dataset used etc. Additionally, it would be useful for them to discuss potential applications or implications of their work beyond data harmonization and domain adaptation.

---

### Official Review · Reviewer_rRrA · 2023-02-03

**Confidence:** 5
**Preliminary Rating:** 2
**Recommendation:** Poster

**Summary:**

The authors design and train a model that predicts image acquisition parameters (IAPs) of MRI slices, such as field strength, echo and repetition times used for the sequence, and contrast agent types, but also more abstract parameters such as the manufacturer of the used scanner. The model achieves accurate predictions, and the authors demonstrate several areas where such a model could be useful in the field.

**Strengths:**

The problem is well-defined and the authors detail various use cases which shows the merit of the paper and the potential value of their proposed model. I agree with the authors that the work is an important step in domain adaptation of medical imaging data.

**Weaknesses:**

I think the model design could be improved to take into consideration the relationship between the IAPs. The variations in the training dataset should also be described in more detail, as in this form, it is hard to tell if the data contained sufficient enough samples.

**Deanonymize Review:**

no

**Detailed Comments:**

- The categorization of the IAPs makes the training of the model more straightforward and the results easy to interpret, however they would hide outlier IAPs not included in these categories (ie. using an image from a new MRI scanner not included in the categories would still be categorized as one of the scanners included). As the model would be used to remove uncertainties and uncover unknown information about the images, it would be interesting to explore some uncertainty metrics for the model, for example Monte-Carlo drop-out. This would perhaps show that although the slice thickness and FOV have lower accuracies, they probably also have larger uncertainties, as---how the authors have also stated,---there are only slight differences between neighboring categories.
- The range of TE and TR values are relatively small. I would recommend including a wider range of values for both training and evaluation.


**Paper Type:**

methodological development

**Questions To Address In The Rebuttal:**

I think the project deals with an important and interesting problem and it is well executed. My main concerns are regarding the presentation of the training dataset, and the issues that might stem from not using a varied enough dataset.
- The authors show how many categories were used to set up the model, but not the distribution of categories in the training and testing datasets. The authors mention possible imbalances in the dataset but don't evaluate it, and I think it holds valuable information for the validity of the model. Please detail if there were any categories (such as any of the possible FOVs) not available for training, and if they were used for evaluating or not.
- Still regarding the variation in training data, although many categories were present, they are most likely very correlated, which raises concerns. The authors discuss that some IAPs can be recovered surprisingly well, such as Field Strength and TE/TR. Without knowing the variations in the training data, I would assume that the images coming from some scanner models generally use the same sequence settings, so if the scanner model is determined, then so are the scanner settings. I believe that some of the high accuracy predictions for difficult problems might come from the fact that the model is looking at an independent but highly correlating parameter.
- Some categories might be irrelevant, as they are included in another independent category. The scanner model always defines the scanner manufacturer, and also the field strength. Having these as three independent outputs adds to the complexity of the model without adding extra value to the predictions. Without knowing more about the data I would assume that the scanner model also correlates well with the patient position, and the acquisition matrix. A better definition of the output IAPs could simplify the model without making it less valuable. Do the authors think that having correlating IAPs as independent outputs shows a better insight into the predictions of the model?
- I believe that the "5.2. Application" part does not add to the significance of the paper, the authors do a great job at describing possible applications for the model, so showing this example might be unnecessary. This space could instead be used on describing the available data in more detail. What do the authors think about the value of this application example in the scope of the paper?
- Summarizing my concerns, I believe the model is trained to solve a much simpler problem than how it is actually designed. Without even considering the continuous TE and TR outputs, the model has 2*8*9*5*2*6*10*21*4*27=195,955,200 possible prediction combinations, but it was trained on only 10,000 image slices, coming from 922 patients, so the possible IAP combinations in the ground truth is limited to 922. This is a problem, since instead of exploiting all the possible variations the presented model could handle, the training dataset imposes a hard limitation about what it can learn. Please provide more information about how many different IAP combinations were present in the training dataset, and if the model was evaluated on any combinations that were not included in training.

---

### Official Review · Reviewer_8RH3 · 2023-02-03

**Confidence:** 5
**Preliminary Rating:** 2
**Recommendation:** Poster

**Summary:**

The authors train a deep learning network to infer acquisition parameters from MRI 2D images. The parameters include scanner type, slice thickness, acquisition resolution, etc…, with both categorical and continuous values. The cost function is therefore a combination of categorical cross-entropy and mean squared error for real values.

**Strengths:**

The system works, with high accuracy in categorical parameters and low errors on real parameters. The application to different models trained on images from different vendors is interesting. The paper is well written and easy to follow. The evaluation is correct.

**Weaknesses:**

While the authors claim that many of the acquisition parameters are not available, it is standard to include them as part of the DICOM headers. For example, patient position, slice thickness, fov, scanner manufacturer and model have their own DICOM fields and are filled by any acquisition device.

**Deanonymize Review:**

no

**Paper Type:**

validation/application paper

**Questions To Address In The Rebuttal:**

It is unclear why the authors decided to model slice thickness, a real-valued parameter, as a categorical variable. Idem for the acquisition matrix, flip angle and FOV computed.

For the testing reported in Table 2, what is the accuracy of the GE Model on GE images? And of the Siemens model in Siemens images? And the accuracy if the images are perfectly classified as being of Siemens or GE?

---

### Official Review · Reviewer_hCCU · 2023-02-04

**Confidence:** 5
**Preliminary Rating:** 4
**Recommendation:** Oral, Poster

**Summary:**

This work proposed a deep learning method for breast DCE MRI image acquisition parameters prediction. Both categorical and continuous acquisition parameters were predicted, most of which with high prediction accuracies. The proposed method was leveraged by a downstream breast cancer classification task to improve classification accuracy.

**Strengths:**

Complete MR acquisition parameters are useful for efficient harmonization/adaptation, so the acquisition parameter prediction task this paper tried to tackle has significant value for the community. Besides, promising prediction accuracies showed potential of the proposed method.

**Weaknesses:**

A major weakness of this paper is that some of the acquisition parameters may not be the appropriate targets to predict, such as “FOV Computed” and  “Acquisition Matrix”, and it is unclear how the image resizing preprocessing affect the acquisition parameters to predict.

**Deanonymize Review:**

no

**Paper Type:**

both

**Questions To Address In The Rebuttal:**

1.	How are images resized and how does the resizing processing affect “Acquisition Matrix” and “FOV Computed”?
2.	Instead of predicting “FOV Computed” and “Acquisition Matrix”, it is more appropriate to predict “In-plane Voxel Size” as the ratio between them, as voxel size is one of the key image parameters affecting downstream task performance. Please make a comment on this.
3.	How to predict slice thickness from 2D slices? Does 3D input improve the accuracy of slice thickness prediction? Some discussion regarding this is preferred.

---

### Official Review · Reviewer_tS8y · 2023-02-04

**Confidence:** 5
**Preliminary Rating:** 1
**Recommendation:** Poster

**Summary:**

The method proposes to predict image acquisition parameters from input MRI data, with the objective of enabling downstream data selection and task forwarding. Methodologically, the paper is water tight, but this does not compensate the limited validation (in the sense that it is only applied to one body part, modality, and site) and the weak premise (IAPs are in the DICOM header).

**Strengths:**

- Good method that clearly performs well on the chosen dataset
- Wide choice of IAPs demonstrate that the method can have broad applicability
- The performance analysis, including looking at different TopK percentages and the tails of the performance distribution, is strong.

**Weaknesses:**

- Method is applied only to 2D slices and using 2D networks. Medical imaging data (and breast MRI) is normally 3D.
- Method is applied only to one body part, modality, and site
- The premise of the model (we do not know the IAPs) is not really a clinically relevant problem (DICOM has this info) or a research problem (as the protocol is defined ahead of time).

**Deanonymize Review:**

no

**Detailed Comments:**

This is a clearly good method that is well developed and performs well on the chosen dataset. The wide choice of IAPs demonstrate that the method can have broad applicability and the overall validation is quite strong, including the part where the authors look at the tails of distributions and system robustness.

However, the paper has major weaknesses, which I expand below:
- Method is applied only to 2D slices and using 2D networks, but medical imaging data, and primarily breast MRI, is normally 3D. This means predictions are done on a per-slice basis, meaning that there will be between-slice prediction differences (which does not make sense for 3D acquisitions). Also, 2D networks are much easier to train due to their size and parameter counts, meaning that the reader is left to guess if they would perform well in 3D.
- Method is applied only to one body part, modality, and site. The title, on the other side, is very broad. One would expect this to be applied to several body regions and different modalities (to test for generalisability), and to different data sources (to test for population and acquisition biases). Without this, the paper is not really of value to many researchers.
- The premise that we do not know the IAPs is not really a clinically relevant problem, as DICOM has this info. In a clinical setting, DICOM images HAVE to have this information otherwise they would be invalid. Authors can argue that this info can be wrong (it happens) but then the training data used in this paper (the target IAPs) might also be wrong. The authors can also argue that this can be applied to a research setting, but again, in a research cohort, the protocol is defined ahead of time. If this is to be applied to random open-source images, then the authors would need to demonstrate that the model generalises to many more sites and data sources.

**Paper Type:**

validation/application paper

**Questions To Address In The Rebuttal:**

There are no questions per se, but simply concerns that are not part of the paper. More specifically:
- Would the method work in 3D? What network would you use and how would you pre-train it?
- Would the method work on other/multiple body regions and modalities?
- Would the method work on data from other sources acquired with different scanners/coils/populations?
- What is the setup where IAPs are actually missing and what is its clinical relevance?

---

### Official Review · Reviewer_4hsH · 2023-02-04

**Confidence:** 3
**Preliminary Rating:** 5
**Recommendation:** Oral, Poster

**Summary:**

This paper proposes a model-based method to solve the inverse problem of recovering the image acquisition parameters (IAPs) that can create an MR image only using the image. Instead of directly predicting the IAPs, the model first predicts 12 attributes that are closely related to IAPs, and then uses these attributes to regress the IAPs. This framework not only can enhance the accuracy of IAPs prediction, but also improve the interpretability of the model. More importantly, the generated MR images significantly benefit cancer classification compared with baseline methods.

**Strengths:**

+ This paper is well written. Motivation, method, experiment, result, and practical impact are clearly described.
+ The results are well presented, not only showing better MR images and IAP predictions, but also improved cancer classification performance.

**Weaknesses:**

No major weakness was detected. Minor concerns are as follows.

- Collecting these 12 attributes requires effort. Are these attributes all important in predicting IAPs? I saw some of the predictions can reach nearly perfect accuracy (Table 1), suggesting that these attributes might be very easy to be determined based on the image. A study on the importance of each attribute to the IAPs prediction is suggested to included (e.g., PCA analysis).

**Deanonymize Review:**

no

**Paper Type:**

methodological development

**Questions To Address In The Rebuttal:**

* A similar idea has been explored in other imaging tasks. For example, to predict tumor benign/malignant, the model first predicts some relevant attributes of the tumor, such as volume, texture features, diagnostic features, etc. Have the authors considered using this information as the input of the model, as the additional information to the image? That is, concatenating these attributes with the image (or image feature) and asking the model to predict the IAPs.

---

### Meta-Review · Area_Chair_jhCz · 2023-02-25

**Recommendation:** Accept (Poster)
**Confidence:** 4

**Metareview:**

This paper received mixed reviews, and I have read all reviews and recommend that it be accepted.